# Lateral/caudal ganglionic eminence makes limited contribution to cortical oligodendrocytes

Jialin Li[1†], Feihong Yang[2†], Yu Tian[1†], Ziwu Wang[1], Dashi Qi[3], Zhengang Yang[1], Jiangang Song[2], Jing Ding[1], Xin Wang[1*], Zhuangzhi Zhang[1*]

[1]State Key Laboratory of Medical Neurobiology and MOE Frontiers Center for Brain Science, Institutes of Brain Science, and Department of Neurology, Zhongshan Hospital, Fudan University, Shanghai, China; [2]Department of Anesthesiology, Shuguang Hospital Affiliated with Shanghai University of Traditional Chinese Medicine, Shanghai, China; [3]Center for Clinical Research and Translational Medicine, Yangpu Hospital, School of Medicine, Tongji University, Shanghai, China

**\*For correspondence:**
wang.xin@zs-hospital.sh.cn (XW);
zz_zhang@fudan.edu.cn (ZZ)

[†]These authors contributed equally to this work

**Competing interest:** The authors declare that no competing interests exist.

**Abstract** The emergence of myelinating oligodendrocytes represents a pivotal developmental milestone in vertebrates, given their capacity to ensheath axons and facilitate the swift conduction of action potentials. It is widely accepted that cortical oligodendrocyte progenitor cells (OPCs) arise from medial ganglionic eminence (MGE), lateral/caudal ganglionic eminence (LGE/CGE), and cortical radial glial cells (RGCs). Here, we used two different fate mapping strategies to challenge the established notion that the LGE generates cortical OPCs. Furthermore, we used a Cre/loxP-dependent exclusion strategy to reveal that the LGE/CGE does not give rise to cortical OPCs. Additionally, we showed that specifically eliminating MGE-derived OPCs leads to a significant reduction of cortical OPCs. Together, our findings indicate that the LGE does not generate cortical OPCs, contrary to previous beliefs. These findings provide a new view of the developmental origins of cortical OPCs and a valuable foundation for future research on both normal development and oligodendrocyte-related disease.

## eLife assessment

The authors provide **solid** evidence that any contribution of oligodendrocyte precursors to the developing cortex from the lateral ganglionic eminence is minimal in scope. The methods used support the conclusions, with some technical concerns that the authors can address with further experimentation. These are considered **valuable** additions to our understanding of the origins of oligodendrocytes in the forebrain during development.

## Introduction

Oligodendrocytes (OLs) are one of the most abundant cell types in the mammalian neocortex. The emergence of myelination OLs in vertebrates is a major event in the evolution of the central nervous system (CNS), as these cells play a crucial role in enhancing axonal insulation and the conduction of neural impulses. Previously, they have been described as support cells that myelinate axons and play essential roles in the function of neocortical circuits. Myelin loss or dysfunction is associated with many disorders, such as multiple sclerosis, depression, and pediatric leukodystrophies (**Goldman et al., 2021**; **Zhou et al., 2021**).

During CNS development, OLs originate from oligodendrocyte precursor cells (OPCs), which arise from different regions of the embryonic neural tube. OPCs proliferate and migrate away from the ventricular germinal zone (VZ) into developing gray and white matter before differentiating into mature OLs (*Miller, 1996*; *Huang et al., 2020*). The developmental origin of cortical OLs has been extensively studied in the CNS, especially in the neocortex (*Allen et al., 2022*; *Liu et al., 2021*; *Shen et al., 2021*; *Marques et al., 2018*). A landmark study *Kessaris et al., 2006* demonstrated that cortical OPCs are generated in three waves, starting with a ventral wave, gradually transitioning to more dorsal origins. The first wave of cortical OPCs is derived from the medial ganglionic eminence (MGE) or the anterior entopeduncular area (AEP) at embryonic day 12.5 (E12.5) (*Nkx2.1*+ lineage). The second wave of cortical OLs is derived from the lateral/caudal ganglionic eminences (LGE/CGE) at E15.5 (*Gsx2*+/*Nkx2.1*- lineage). The final wave occurs at P0, when OPCs originate from cortical glial progenitor cells (*Emx1*+ lineage). Our recent studies (*Li et al., 2021*; *Zhang et al., 2020*), however, challenge the notion that cortical OLs are generated in these three waves (*Kessaris et al., 2006*; *Richardson et al., 2006*; *Rowitch and Kriegstein, 2010*; *Bergles and Richardson, 2015*), as the cortical glial progenitor cells were also found to express *Gsx2*, which was known to be exclusively expressed in the developing subpallium (*Chapman et al., 2018*; *Chapman et al., 2013*; *Wen et al., 2021*), after E16.5 (*Zhang et al., 2020*). Therefore, *Gsx2*+/*Nkx2.1*- lineage OLs may not represent LGE-derived cortical OLs. To date, whether the LGE/CGE generates cortical OPCs (OLs) during cortical development remains an unanswered question in the field of developmental neuroscience.

The present study was undertaken to answer this question. We carried out distinct fate mapping experiments and intersectional lineage analyses to reveal that cortical OPCs do not derive from the LGE/CGE. First, using in utero electroporation (IUE) to target the LGE in a Cre recombinase-dependent IS (*Rosa26*CAG-LSLFrt-tdTomato-Frt-EGFP) reporter mouse line (*He et al., 2016*), we found that LGE-derived OPCs produced striatal OLs rather than cortical OLs. Second, by combining the PiggyBac transposon system with IUE, we found that no traced cells migrated into the cortex from the LGE. Third, by crossing the *Nkx2.1*Cre and *Emx1*Cre mouse lines with an H2B-GFP reporter mouse line, we found that nearly all cortical OPCs originated from *Nkx2.1*Cre or *Emx1*Cre lineage cells. Finally, we analyzed *Olig2* conditional knockout mice (*Nkx2.1*Cre; *Olig2*F/F mice), and found that the number of cortical OPCs was significantly reduced from E14.5 to E16.5. These results indicated that the MGE was the predominant ventral source of cortical OPCs. Taken together, our findings suggest that the LGE/CGE gives rise to limited cortical OPCs (OLs) during neocortical development.

## Results

### Fate mapping of LGE-derived OPCs by combining IUE with a Cre recombinase-dependent IS reporter

OPCs initially emerge in the ventral VZ of the MGE and can be labeled by platelet-derived growth factor receptor (α-subunit, *Pdgfra*) around E12.5. We hypothesized that if the LGE generated cortical OPCs, the *Pdgfra*+ cell migration stream from the LGE to the cortex could be observed during cortical development. To confirm this speculation, we conducted a detailed analysis of *Pdgfra* mRNA expression using in situ hybridization (ISH). Our results showed that there was no observable *Pdgfra*+ cell migration stream from the LGE to the cortex from E13.5 to postnatal day 0 (P0) (*Figure 1A*). Next, we used Cre recombinases in combination with an IS reporter mouse line (*He et al., 2016*) to fate map LGE-derived OPCs (OLs). We delivered pCAG-Cre plasmids specifically into the LGE VZ of IS reporter mouse embryos at E13.5 by IUE (*Figure 1B and C*). In this experiment, cells generated from electroporated neural stem cells (NSCs) expressed tdTomato (tdT). Interestingly, we observed many tdT+ cells throughout the whole LGE (striatum) at E18.5. In contrast, there were very few tdT+ cells detected in the cortex (*Figure 1D*). OPCs express *Olig2*, *Sox10*, *Pdgfra*, and NG2 (*Cspg4*) and mature neurons express NeuN (*Sun et al., 2001*; *Zhou et al., 2001*; *Lu et al., 2002*; *Finzsch et al., 2008*; *Reiprich et al., 2017*; *Kucharova and Stallcup, 2010*; *Sánchez-González et al., 2020*). To determine the identity of the tdT+ cells, we performed double staining of tdT with OLIG2, SOX10, PDGFRA, NG2 (CSPG4), and NeuN. Within the cortex, tdT+ cells did not express OPC marker genes but express NeuN; however, tdT+ cells expressed OPC marker genes and NeuN in the striatum (*Figure 1E and F*). We also delivered pCAG-Cre plasmids specifically into the LGE VZ of IS reporter mice at P0 by IUE and analyzed these mice at P49 (*Figure 1G*) to investigate whether the LGE generates cortical OLs

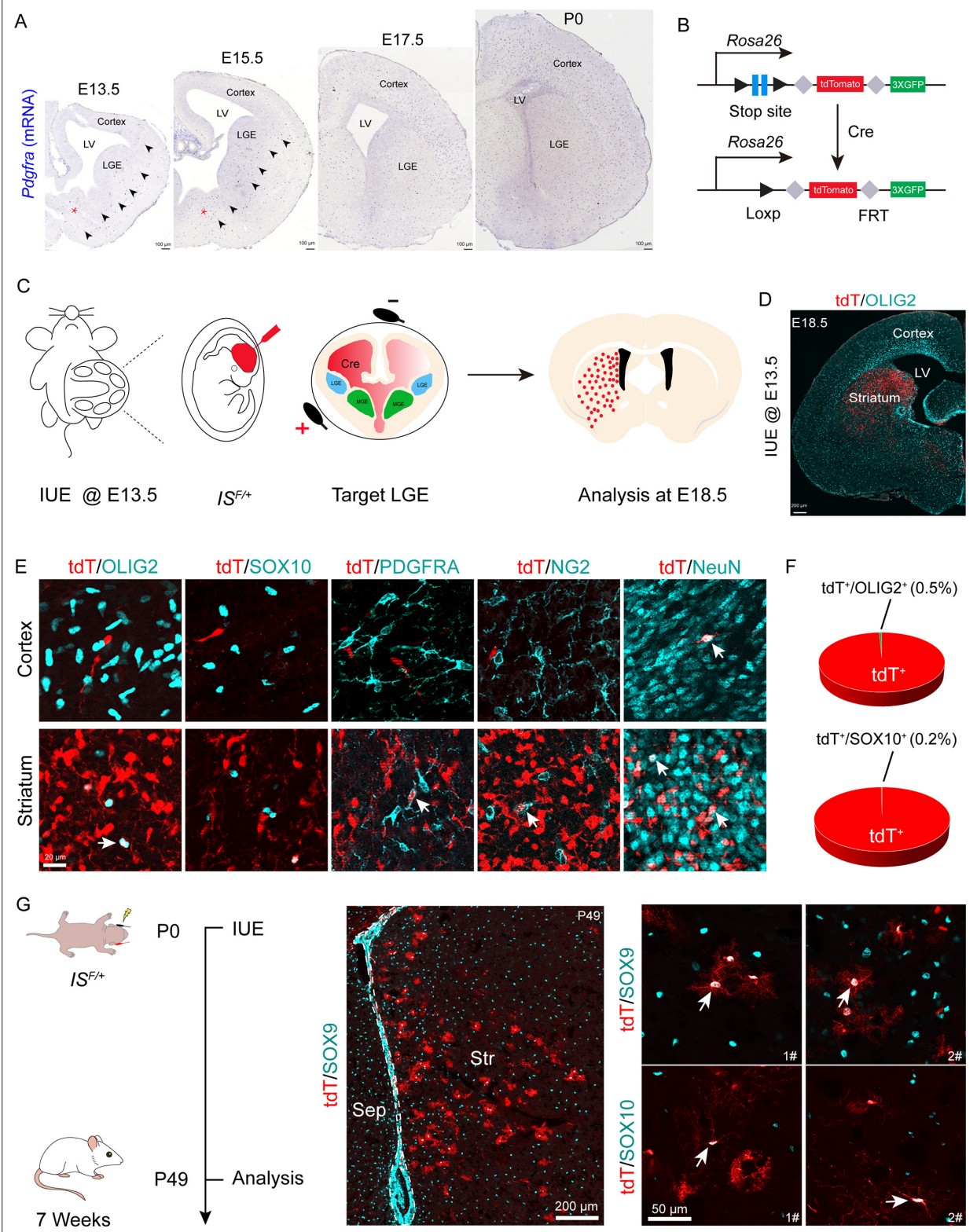

**Figure 1.** Fate mapping of lateral ganglionic eminence (LGE)-derived oligodendrocyte precursor cells (OPCs) by combining in utero electroporation (IUE) with a Cre recombinase-dependent IS reporter. (**A**) In situ hybridization showing *Pdgfra* expression from embryonic day 13.5 (E13.5) to postnatal day 0 (P0). Arrowheads indicated the *Pdgfra*+ cell migration stream from the medial ganglionic eminence (MGE)/anterior entopeduncular area (AEP) to the cortex. (**B**) Scheme of the IS reporter lines. (**C**) Experimental schedule to trace LGE-derived OPCs. (**D**) Representative coronal sections showing the distribution of tdT+ cells. (**E**) tdT+ cells expressed NeuN but not OLIG2, SOX10, PDGFRA, or NG2 in the cortex. In contrast, the tdT+ cells expressed all of

Figure 1 continued

these markers (OLIG2, SOX10, PDGFRA, NG2, and NeuN) in the striatum. (**F**) The ratio of the OLIG2$^+$ and SOX10$^+$ cells of the electroporated cells in the cortex. N=4 mice per group. (**G**) Fate mapping of LGE-derived cells at P0. tdT$^+$ cells expressed SOX10 and SOX9 in the striatum at P49.

The online version of this article includes the following source data and figure supplement(s) for figure 1:

**Source data 1.** The raw data for the visualization of data presented in *Figure 1F*.

**Figure supplement 1.** Lineage tracing of lateral ganglionic eminence (LGE)-derived oligodendrocyte precursor cells (OPCs) by combining in utero electroporation (IUE) with a Cre recombinase-dependent IS reporter.

at a later developmental stage. Our results showed that LGE progenitors generate striatum OLs and astrocytes, which expressed SOX10 and SOX9, respectively (*Figure 1G*).

To further determine whether LGE-derived OPCs migrate into the cortex after E18.5, we used the same strategy as mentioned above (*Figure 1—figure supplement 1A*). We observed that at P10, the majority of the tdT$^+$ cells were located in the striatum, whereas very few tdT$^+$ cells were located in the neocortex (*Figure 1—figure supplement 1B*). All tdT$^+$ cells in the neocortex expressed NeuN but did not express OLIG2 (*Figure 1—figure supplement 1C*). In fact, tdT$^+$ cells in the neocortex did not express OPC marker genes; instead, they expressed NeuN in the neocortex. In contrast, tdT$^+$ cells in the striatum expressed OPC marker genes (*Olig2*, *Sox10*, and *Pdgfra*) and NeuN (*Figure 1—figure supplement 1C and D*), consistent with our observation at E18.5 (*Figure 1E*). We also found that tdT$^+$ cells expressed the astrocyte marker ALDH1L1 in the striatum at P10 (*Figure 1—figure supplement 1D*). Furthermore, through consecutive tissue sectioning and 3D reconstruction, we revealed that the traced cells tended to be localized in the striatum (*Figure 1—figure supplement 1E*). Collectively, our findings indicated that the LGE gives rise to striatal OLs, neurons, and astrocytes, but not cortical OLs.

## Lineage tracing of LGE-derived OPCs by combining IUE with the PiggyBac transposon system

To further investigate whether the LGE generates cortical OPCs during cortical development, we employed an alternative fate mapping strategy by combining IUE with the PiggyBac transposon system to map the fate of OPCs derived from LGE. This approach utilized a transposase enzyme (helper plasmid) to integrate the H2B-GFP sequence from a donor plasmid into the genome of the in utero electroporated LGE progenitors along the lateral ventricular zone. The H2B-GFP reporter vector was delivered specifically into the LGE VZ of wild-type embryos at E13.5 by IUE, and these embryos were examined at E18.5 or P10 (*Figure 2A and B*). In this system, most traced cells (GFP$^+$) were located in the striatum, and very few traced cells were observed in the cortex (*Figure 2C and E*). Usually, a small population of RGCs located in the dorsal LGE or lateral/ventral pallium was labeled by IUE. These RGCs mainly migrated to the lateral/ventral pallium and amygdala (*Figure 2C*). We did not observe any GFP$^+$ cells that were OPCs in the dorsal cortex, as these cells did not express OLIG2, SOX10, or PDGFRA (*Figure 2D*). Instead, they expressed NeuN (*Figure 2D*). In contrast, GFP$^+$ cells expressed OLIG2, SOX10, PDGFRA, and NeuN in the striatum at E18.5 (*Figure 2D*). The same results were also observed at P10 (*Figure 2E*). Again, our findings strongly suggested that the LGE may not generate cortical OPCs.

## An exclusion strategy revealed that LGE/CGE contribution to cortical OPCs is minimal

A previous report (*Kessaris et al., 2006*) suggested that cortical OPCs originate from MGE, LGE/CGE, and cortical RGCs. When MGE-derived cortical OPCs were eliminated, LGE/CGE-derived cells accounted for approximately 50% of cortical OPCs at P10. Subsequently, another study reported that approximately 20% OLs in corpus callosum are ventral-derived at P12/P13 (*Tripathi et al., 2011*). However, according to our previous findings (*Li et al., 2021*; *Zhang et al., 2020*), we speculate that these OPCs (*Gsx2*$^+$ lineage) may derived from the cortical progenitors (*Emx1*$^+$ lineage). To confirm these findings, using the *Emx1*$^{Cre}$; H2B-GFP mice, we found that many GSX2$^+$ cells were located in the cortical SVZ. Interestingly, these GSX2$^+$ cells were co-labeled with GFP (*Emx1*$^+$ lineage), suggesting that the cortical SVZ GSX2$^+$ cells were derived from the cortical progenitors (*Figure 3—figure supplement 1A–C*).

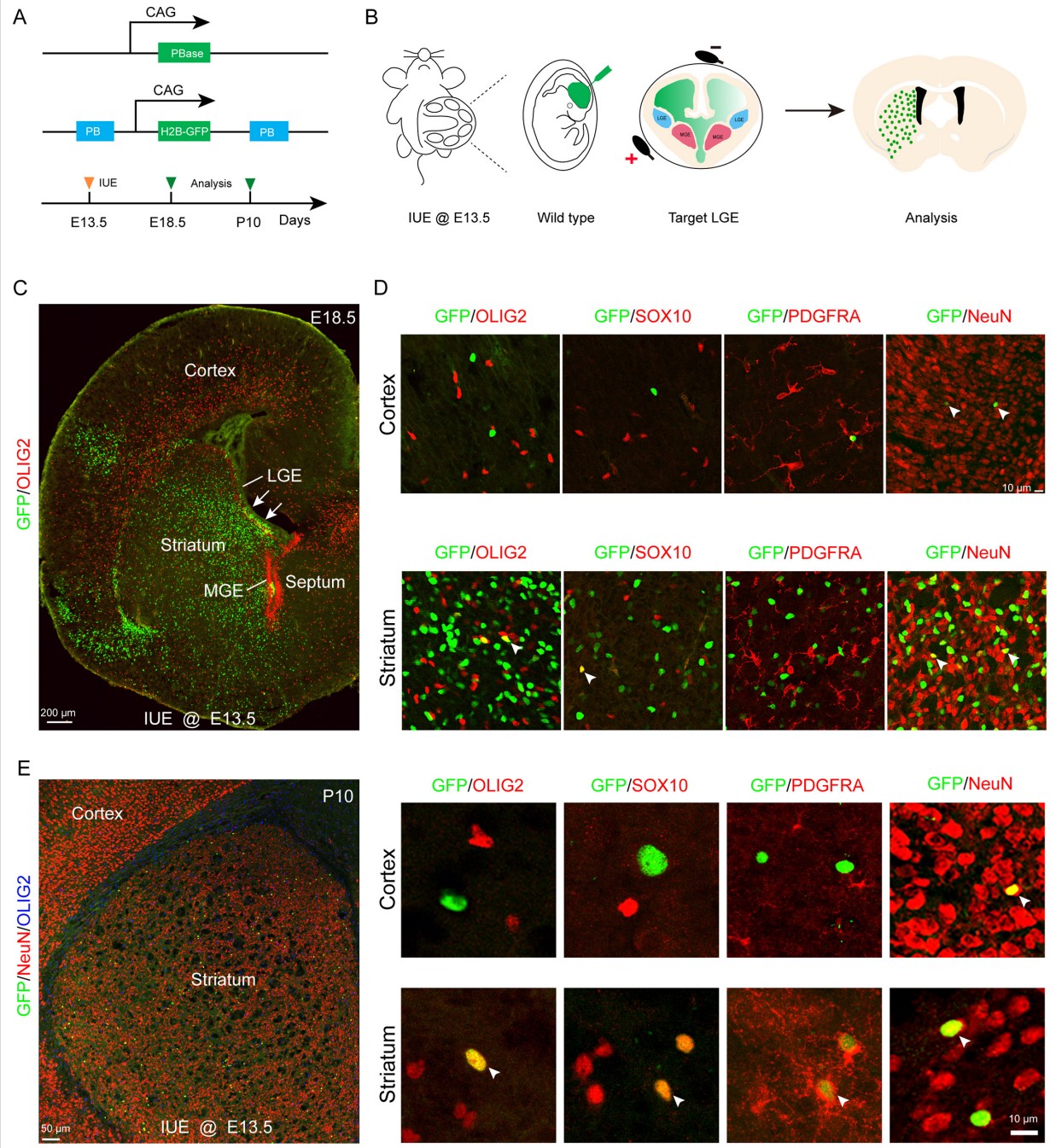

**Figure 2.** Fate mapping of lateral ganglionic eminence (LGE)-derived oligodendrocyte precursor cells (OPCs) by combining in utero electroporation (IUE) with PiggyBac transposon system. (**A**) Schematic of the PiggyBac transposon reporter system used in this study. (**B**) The experimental workflow. (**C**) Representative coronal sections showing the distribution of GFP+ cells at embryonic day 18.5 (E18.5). (**D**) GFP+ cells expressed NeuN but not glial cell markers, such as OLIG2, SOX10, and PDGFRA in the cortex. However, GFP+ cells expressed NeuN, OLIG2, SOX10, and PDGFRA in the striatum. (**E**) GFP+ cells expressed NeuN but not glial cell markers, such as OLIG2, SOX10, and PDGFRA in the cortex and GFP+ cells expressed NeuN, OLIG2, SOX10, and PDGFRA in the striatum at postnatal day 10 (P10). N=4 per group.

While distinct fate mapping strategies provided evidence that the LGE may not generate cortical OPCs during cortical development, there remained a possibility that a small population of LGE/CGE RGCs or progenitors might generate a subpopulation of cortical OPCs at specific time points. To address this possibility, we crossed the *Emx1^Cre* (***Gorski et al., 2002***; ***Du et al., 2022***; ***Li et al., 2024***), *Nkx2.1^Cre* (***Xu et al., 2008***), and H2B-GFP (***He et al., 2016***) mouse lines to generate *Emx1^Cre*;

*Nkx2.1^Cre*; H2B-GFP mice. Because *Emx1^Cre* was specifically expressed in cortical RGCs (*Gorski et al., 2002*) and *Nkx2.1^Cre* was specifically expressed in MGE/AEP RGCs (*Xu et al., 2008*), cortical OPCs were traced (GFP⁺ cells) in using *Emx1^Cre*; *Nkx2.1^Cre*; H2B-GFP mice. GFP⁻ cell OPCs, on the other hand, were considered as LGE/CGE-derived cortical OPCs (OLs) (*Figure 3A and B*). We performed double staining of GFP with OPC marker genes, such as *Sox10* and *Pdgfra* in *Emx1^Cre*; *Nkx2.1^Cre*; H2B-GFP mice (*Figure 3C*). We observed that the majority of OPCs or OLs were co-labeled with GFP in *Emx1^Cre*; *Nkx2.1^Cre*; H2B-GFP mice at P10 (*Figure 3C*). Statistical analysis showed that only approximately 2% of OPCs were GFP⁻ cells, and they were possibly derived from the LGE/CGE (*Figure 3D and E*), which was consistent with our observation at P0 (*Figure 3G–I*). We also found that many GFP⁺ cells expressed SOX10 and PDGFRA in the striatum at P10 (*Figure 3F*). Importantly, the expression of *Nkx2.1^Cre* is lower within the dorsalmost region of the MGE than in other *Nkx2.1*-expressing regions (*Xu et al., 2008*). Thus, these GFP⁻ OPCs (~2%) may be derived from the dorsal MGE rather than from the LGE/CGE. Using this approach, we revealed that nearly all cortical OLs/OPCs were derived from MGE/AEP and cortical RGCs. Taken together, our findings indicated that the contribution of cortical OPCs from the LGE/CGE is minimal.

## The MGE/AEP was the predominant ventral source of the cortical OPCs

*Olig2*, a basic helix-loop-helix transcription factor, plays a critical role in the regulation of OPC fate determination. To investigate the role of *Olig2* in the MGE, we crossed *Nkx2.1^Cre* mice with *Olig2^F/+* mice to eliminate *Olig2* expression in the SVZ/VZ of the MGE/AEP (*Figure 4A and B*). We referred to the conditional mutants as *Olig2*-NCKO mice. A previous study (*Kessaris et al., 2006*) showed that MGE/AEP-derived OPCs migrate into the cortex around E14.5, and LGE/CGE-derived OPCs migrate into cortex around E15.5. We assumed that the MGE/AEP was the predominant ventral source of cortical OPCs, and that inhibiting MGE/AEP-derived OPC generation would therefore significantly reduce the number of cortical OPCs at E16.5. To test this possibility, we analyzed *Olig2*-NCKO mice, in which the generation of MGE/AEP-derived OPCs was inhibited. Our results showed that the expression of OPC marker genes, such as *Sox10* and *Pdgfra,* was significantly reduced (>10-fold) in the cortex of *Olig2*-NCKO mice, compared to the control mice at E14.5 and E16.5 (*Figure 4C–E*). Considering the recombination efficiency of the CRE enzyme, the limited presence of residual OPCs in the cortex of *Olig2*-NCKO mice may be arise from the MGE/AEP. Indeed, we observed few PDGFRA-positive cells in the MGE/AEP of the *Olig2*-NCKO mice (*Figure 4F*). Altogether, our findings indicated that the MGE/AEP was the predominant ventral source of cortical OPCs.

## Discussion

It is widely accepted in the field of developmental neuroscience that cortical OPCs originate from three regions: the MGE/AEP, LGE/CGE, and cortex. However, in the current study, we re-examined the developmental origins of cortical OPCs using diverse strategies. Our findings show that cortical OPCs do not come from the LGE/CGE, contrary to previous reports (*Kessaris et al., 2006*; *Richardson et al., 2006*), which suggested that the LGE/CGE generates cortical OPCs (*Figure 4G*). In fact, no cortical OPCs migrating from the LGE to the cortex were observed via our two fate mapping strategies. Furthermore, an exclusion strategy suggested that the LGE/CGE were not responsible for generating cortical OPCs. In addition, according to our study of *Olig2*-NCKO mice, MGE/AEP seems to be the predominant ventral source of cortical OPCs during neocortical development. Overall, these findings provide strong evidence that the contribution of the cortical OPCs from the LGE/CGE is minimal in the mouse brain.

## Why LGE is not the main OPC pool for the cortex during development

OLs are specialized cells of the CNS that produce myelin, a multilayered membrane that spirally enwraps axons and facilitates rapid nerve conduction. Recent studies have shown that cortical OLs are heterogeneous, as there are transcriptionally distinct subpopulations of cortical OLs during development and disease (*Marques et al., 2018*; *Floriddia et al., 2020*; *Marques et al., 2016*; *Falcão et al., 2018*). Indeed, OLs are derived from distinct OPC pools, which acquire different abilities to respond to demyelination (*Kessaris et al., 2006*; *Crawford et al., 2016*). In the spinal cord, the developmental origin of OLs has been hotly debated for many years. Some groups believed that OLs originate only

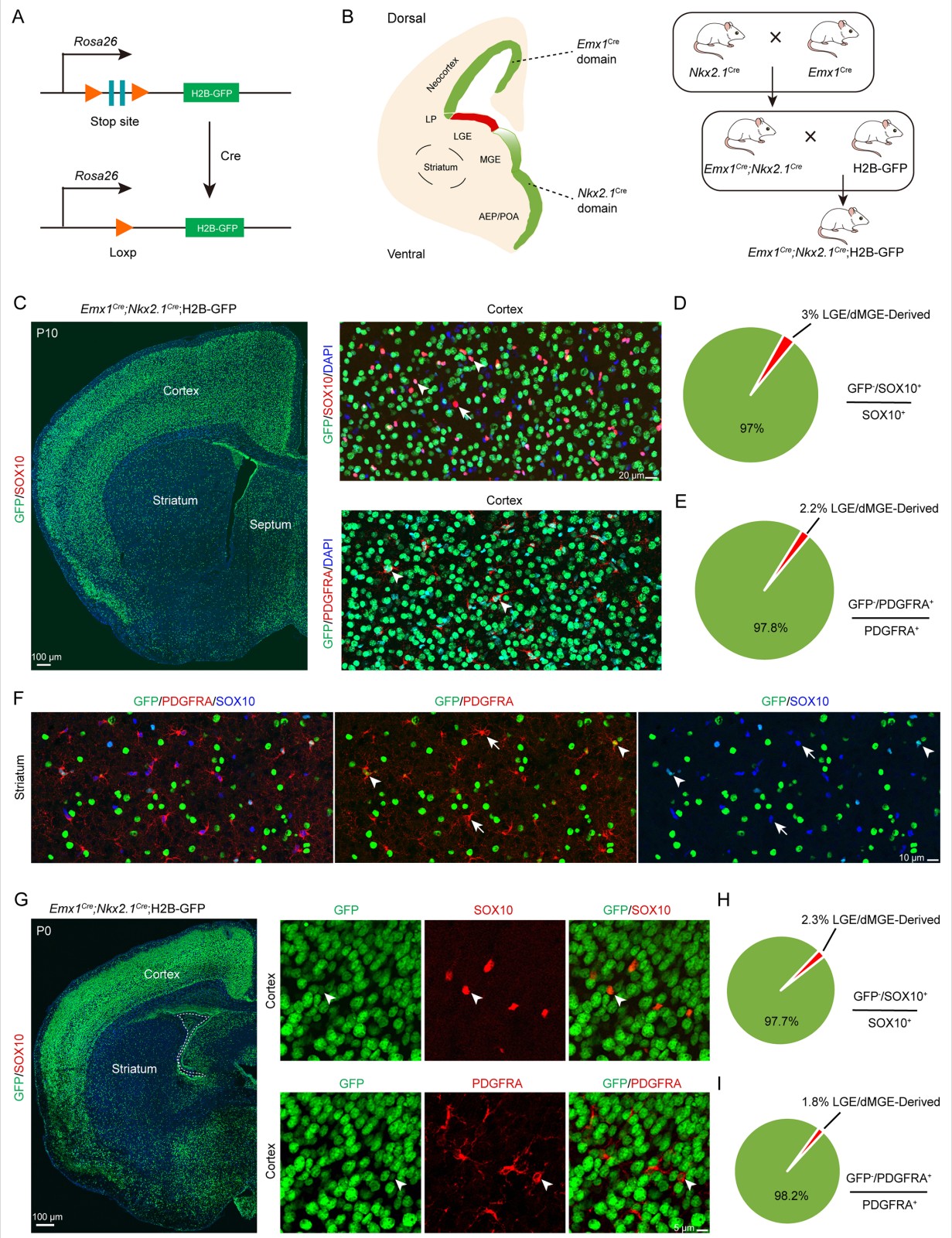

**Figure 3.** An exclusion strategy showed that lateral/caudal ganglionic eminence (LGE/CGE) contribution to cortical oligodendrocyte precursor cells (OPCs) is minimal. (**A**) Scheme of the H2B-GFP reporter lines. (**B**) Experimental design of the exclusion strategy to trace the lineage of LGE radial glial cells (RGCs). (**C**) Representative coronal sections showing the traced cells in the forebrain. The majority of SOX10- and PDGFRA-positive cells were GFP⁺ cells in the cortex at postnatal day 10 (P10). (**D–E**) The pie chart shows that the percentage of LGE/dMGE-derived cortical OPCs was approximately 3%.

*Figure 3 continued on next page*

*Figure 3 continued*

N=6 mice per group. (**F**) Many GFP[+] cells expressed SOX10 and PDGFRA in the striatum at P10. (**G**) Nearly all SOX10 and PDGFRA expressed GFP in the cortex at P0. (**H–I**) The pie chart shows that the percentage of LGE/dMGE-derived cortical OPCs was less than 3%. N=6 mice per group.

The online version of this article includes the following source data and figure supplement(s) for figure 3:

**Source data 1.** The raw data for the visualization of data presented in *Figure 3D, E, H, I*.

**Figure supplement 1.** Lineage tracing of *Emx1*[Cre] derived cortical cells.

**Figure supplement 1—source data 1.** The raw data for the visualization of data presented in *Figure 3—figure supplement 1C*.

from the ventral neural tube, whereas others tend to support multiple origins (*Richardson et al., 2006*; *Yu et al., 1994*; *Spassky et al., 1998*; *Fogarty et al., 2005*). One pioneering study (*Kessaris et al., 2006*) purported that, in the forebrain, cortical OLs derive from multiple origins, such as the MGE, LGE, and cortex.

In general, the specification of OLs depends on SHH protein released from the notochord and floor plate, as *Shh* expression is both necessary and sufficient for OL generation in the spinal cord (*Orentas et al., 1999*). As in the spinal cord, OL generation in the cortex is also dependent on the *Shh* signaling pathway (*Lu et al., 2000*). During MGE development, *Nkx2.1* and *Six3* promote SHH secretion. Furthermore, the early-born neurons of the MGE also secrete SHH protein before E14.5 (*Flandin et al., 2011*). Thus, the high levels of SHH in the MGE contribute to the generation of OPCs from RGCs. In contrast, *Shh* signaling pathway activity is very weak in the LGE at the early stage (before E14.5). As the level of SHH gradually increases in the LGE, this results in a shift from neurogenesis to gliogenesis after E15.5, and the large number of the striatal neurons may block the migration of OPCs into the neocortex (*Tong et al., 2015*; *Su et al., 2022*). In summary, the process of cortical OL generation can be summarized as follows: First, the high level of SHH in the MGE allows for the production of a small population of cortical OPCs around E12.5. Subsequently, multipotent intermediate progenitors begin to express DLX transcription factors resulting in ending the generation of OPCs in the MGE (*Dai et al., 2015*; *Silbereis et al., 2014*; *Price et al., 2022*). Second, RGCs undergo a gradual transition from neurogenesis to gliogenesis around E15.5, driven by the progressive elevation of SHH levels in the LGE. However, the LGE had already generated a substantial number of neurons during this period, blocking the migration of OPCs into the cortex. Finally, with the gradual increase of SHH concentration, cortical RGCs begin to cease neurogenesis and shift toward gliogenesis after E16.5 (*Li et al., 2021*; *Tong et al., 2015*; *Winkler et al., 2018*). Therefore, the generation of cortical OPCs is regulated by both precise temporal and spatial patterning, which is strictly dependent on the local environment.

## Cortical-derived OLs/OPCs make up the majority of the cortical OLs/OPCs in the adult brain

During the early development of the cortex, both pallium and subpallium can generate cortical OPCs (*Kessaris et al., 2006*; *van Tilborg et al., 2018*). Interestingly, most of the ventral-derived OPCs, particularly MGE-derived OPCs/OLs, are eventually eliminated due to the continued expansion of dorsal-derived OPCs in the adult cortex (*Liu et al., 2021*; *Kessaris et al., 2006*). It appears that OPCs generated later in development may possess a competitive advantage over their earlier-generated counterparts. Indeed, human glial progenitor cells engrafted neonatally or transplanted into the adult mouse brain can effectively disperse throughout the forebrain, differentiate into mature OLs, and remyelinate the demyelinated adult brain (*Goldman et al., 2021*; *Windrem et al., 2020*). These findings indicate that OPCs generated later in development have the ability to take over for earlier-generated OPCs in the adult brain.

Based on the present work and other studies (*Marques et al., 2018*; *Kessaris et al., 2006*; *van Tilborg et al., 2018*), cortical-derived OPCs probably constitute the majority of cortical OPCs, including mature cortical OLs. Thus, OLs may function as a homogeneous population in the adult forebrain. In fact, the number of LGE/CGE-derived cortical OPCs may also gradually decrease in the adult brain, as reported by *Liu et al., 2021*. However, Lui et al. did not analyze LGE-derived and MGE-derived OPCs separately. Therefore, they did not exclude the possibility that the LGE/CGE does not generate cortical OPCs. In summary, cortical OPCs/OLs comprise MGE-derived and cortical RGC-derived OPCs/OLs during the neonatal stage. In contrast, cortical-derived OPCs/OLs make up

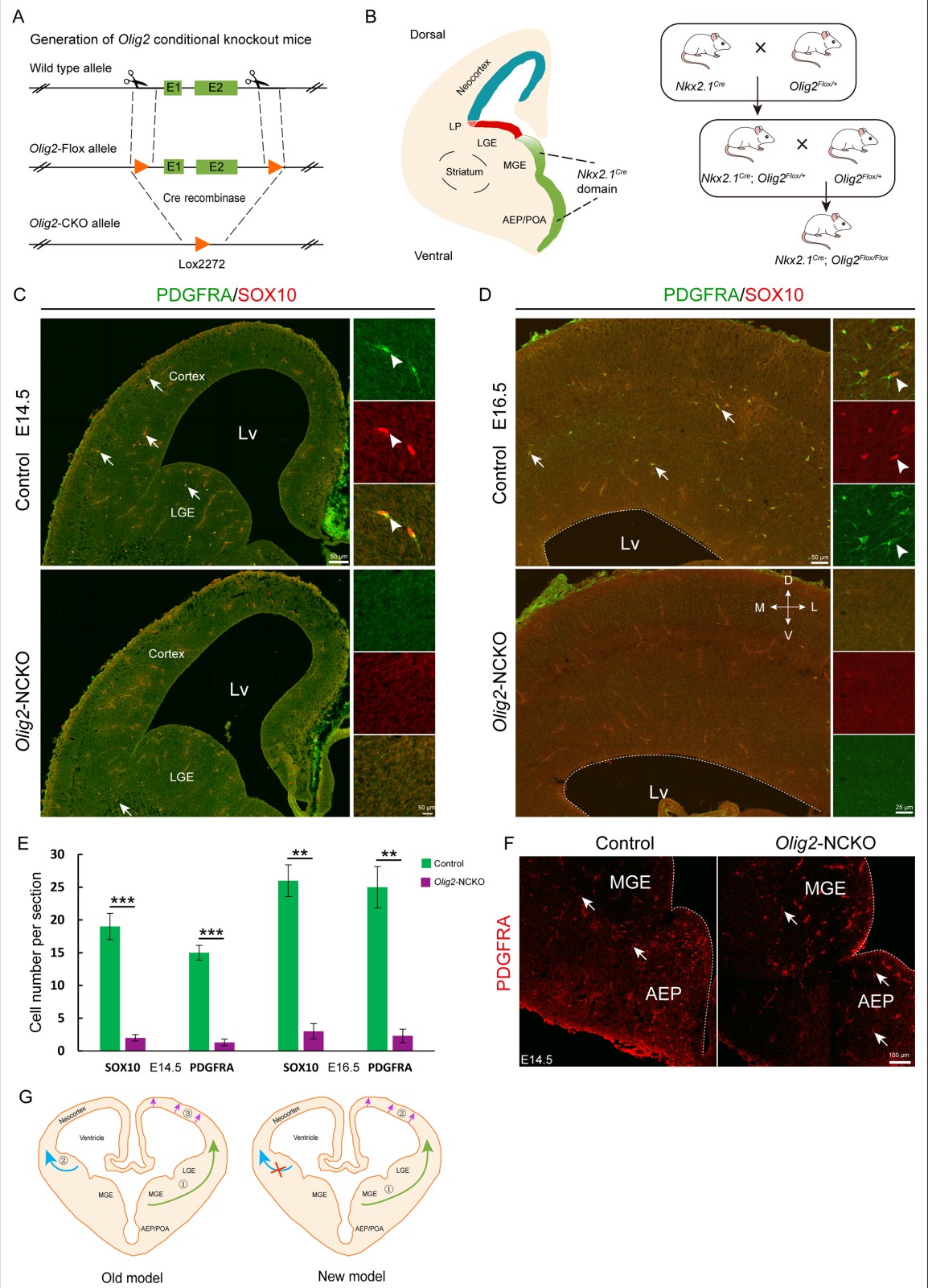

**Figure 4.** The medial ganglionic eminence (MGE) may be the sole ventral source of cortical oligodendrocyte precursor cells (OPCs). (**A**) The CRISPR/Cas9 technique was used to generate an *Olig2* conditional knockout allele. (**B**) Experimental design for the generation of *Olig2*-NCKO mice. (**C–D**) The expression of SOX10 and PDGFRA was significantly reduced in the cortex of *Olig2*-NCKO mice compared with that of control mice. (**E**) The number of SOX10- and PDGFRA-positive cells was significantly reduced in *Olig2*-NCKO mice compared with control mice. Student's t-test, **p<.01, ***p<.001,

*Figure 4 continued on next page*

the majority of the cortical OPCs/OLs in the adult cortex. Recently, another study showed that ventral-derived OLs contribution to cortex is minimal.

Our study provides strong evidence that the contribution of the LGE to cortical OPCs is minimal during forebrain development. Thus, our findings provide new evidence about the developmental origins of cortical OPCs, which may guide future research on both normal development and disease.

# Materials and methods

**Key resources table**

| Reagent type (species) or resource | Designation | Source or reference | Identifiers | Additional information |
|---|---|---|---|---|
| Genetic reagent (*Mus musculus*) | *Emx1Cre* | The Jackson Laboratory | Stock No. 005628 | |
| Genetic reagent (*Mus musculus*) | *Nkx2.1Cre* | The Jackson Laboratory | Stock No. 008661 | |
| Genetic reagent (*Mus musculus*) | IS reporter | The Jackson Laboratory | Stock No. 028582 | |
| Genetic reagent (*Mus musculus*) | *Olig2F/+* | Our lab | N/A; available from the authors | |
| Genetic reagent (*Mus musculus*) | C57BL/6 | Department of Laboratory Animal Science at Fudan University | http://10.107.12.196/ | |
| Genetic reagent (*Mus musculus*) | ICR | Department of Laboratory Animal Science at Fudan University | http://10.107.12.196/ | |
| Antibody | Anti-GFP (Chicken polyclonal) | Aves Labs | Cat# GFP-1020, RRID: AB_10000240 | IF (1:3000) |
| Antibody | Anti-ALDH1L1 (Rabbit polyclonal) | Abcam | Cat# ab87117, RRID: AB_10712968 | IF (1:1000) |
| Antibody | Anti-OLIG2 (Rabbit polyclonal) | Oasis Biofarm | Cat# OB-PRB009, RRID: AB_2934240 | IF (1:500) |
| Antibody | Anti-OLIG2 (Rat polyclonal) | Oasis Biofarm | Cat# OB-PRT020, RRID: AB_2934241 | IF (1:500) |
| Antibody | Anti-OLIG2 (Mouse polyclonal) | Millipore | Cat# MABN50, RRID: AB_10807410 | IF (1:1000) |
| Antibody | Anti-OLIG2 (Rabbit polyclonal) | Millipore | Cat# AB9610, RRID: AB_570666 | IF (1:1000) |
| Antibody | Anti-SOX10 (Goat polyclonal) | R&D Systems | Cat# ab442208, RRID: unknown | IF (1:500) |
| Antibody | Anti-SOX10 (Guinea Pig polyclonal) | Oasis Biofarm | Cat# OB-PGP001, RRID: AB_2934230 | IF (1:1000) |
| Antibody | Anti-PDGFRA (Rabbit polyclonal) | Oasis Biofarm | Cat# OB-PRB051, RRID: AB_2938684 | IF (1:800) |
| Antibody | Anti-NeuN (Rabbit unknown) | Bioscience | Cat# R-3770–100, RRID: AB_2493045 | IF (1:1000) |

*Continued on next page*

*Continued*

| Reagent type (species) or resource | Designation | Source or reference | Identifiers | Additional information |
|---|---|---|---|---|
| Antibody | Anti-NeuN (Rabbit polyclonal) | Oasis Biofarm | Cat# OB-PRB039, RRID: AB_2934232 | IF (1:1000) |
| Antibody | Anti-NG2 (Guinea Pig polyclonal) | Oasis Biofarm | Cat# OB-PGP002, RRID: AB_2938678 | IF (1:1000) |
| Antibody | Alexa488-Conjugated Affinipure Donkey Anti-Chicken IgY++ (IgG) (H+L) | Jackson ImmunoResearch Labs | Cat# 703-545-155, RRID: AB_2340375 | IF (1:500) |
| Antibody | CyTM3-Conjugated Affinipure Donkey Anti-Rabbit IgG (H+L) | Jackson ImmunoResearch Labs | Cat# 711-165-152, RRID: AB_2307443 | IF (1:500) |
| Antibody | Alexa647-Conjugated Affinipure Donkey Anti-Rat IgG (H+L) | Jackson ImmunoResearch Labs | Cat# 712-605-153, RRID: AB_2340694 | IF (1:500) |
| Antibody | Alexa647-Conjugated Affinipure Donkey Anti-Goat IgG (H+L) | Jackson ImmunoResearch Labs | Cat# 705-605-147, RRID: AB_2340437 | IF (1:500) |
| Antibody | Alexa488-Conjugated Affinipure Donkey Anti-Rabbit IgG (H+L) | Jackson ImmunoResearch Labs | Cat# 711-545-152, RRID: AB_2313584 | IF (1:500) |
| Antibody | Cy3-AffiniPure Donkey Anti-Rat IgG (H+L) | Jackson ImmunoResearch Labs | Cat# 712-165-150, RRID: AB_2340666 | IF (1:500) |
| Antibody | Cy3-AffiniPure Donkey Anti-Guinea Pig IgG (H+L) | Jackson ImmunoResearch Labs | Cat# 706-165-148, RRID: AB_2340460 | IF (1:500) |
| Recombinant DNA reagent | pCAGIG (plasmid) | Addgene | Cat# 11159 | |
| Recombinant DNA reagent | pGL4.10 (plasmid) | Promega | Cat# E6651 | |
| Recombinant DNA reagent | pGL4.23 (plasmid) | Promega | Cat# E8411 | |

## Animals

All experiments conducted in this study were in accordance with the guidelines of Fudan University (No. 20220228-140). We used *Emx1^Cre^* (*Gorski et al., 2002*; *Du et al., 2022*) and *Nkx2.1^Cre^* (*Xu et al., 2008*) mice, in conjunction with either *Rosa26-H2B-GFP* (*He et al., 2016*) or *Rosa26-IS* (*He et al., 2016*) for genetic fate mapping. *Olig2^F/+^* mice were generated via the CRISPR/Cas9 strategy. LoxP sites flanked the coding region of exons 1 and 2. Wild-type mice or littermates without the Cre allele were used as controls. All mice were maintained on a mixed genetic background (C57BL/6J and CD1). The mice were allowed access to water and food ad libitum and maintained on a 12 hr light/dark cycle. The day of vaginal plug detection was considered E0.5, and the day of birth was defined as P0. Both male and female mice were used in all experiments.

## Tissue preparation

Postnatal mice were deeply anesthetized and perfused intracardially thoroughly with ice-cold PBS followed by 4% paraformaldehyde (PFA). Embryos were harvested from deeply anesthetized pregnant mice. All brains were fixed overnight in 4% PFA at 4°C, cryoprotected in 30% sucrose for at least 24 hr, embedded in optimal cutting temperature (Sakura Finetek) on dry ice and ethanol slush and preserved at –80°C. All frozen mouse tissues were sectioned into 20 µm thick sections unless specifically mentioned and stained on glass slides.

## Plasmid construction

The pCAG-Cre vector was constructed as reported elsewhere. The coding sequence of Cre in the pCAG-Cre vector was replaced by the PiggyBac transposase to construct the pCAG-PiggyBac transposase vector. The coding DNA sequence of *Cre* in the pCAG-Cre vector was replaced by H2B-GFP from the *Rosa26*-H2B-GFP mouse strain to construct the PB-pCAG-H2B-GFP vector. The inverted

terminal repeat sequence of the PiggyBac transposon was cloned upstream of the *CAG* promoter and downstream of the beta-globin polyA sequence for recognition by the PiggyBac transposase.

## In utero electroporation

IUE of wild-type or IS embryos was performed at E13.5. The pCAG-Cre (Addgene #13775), pCAG-GFP (Addgene #11150), pCAG-PiggyBac transposase, or PB-pCAG-H2B-GFP (final concentration: 1–2 µg/µL) plasmid was mixed with 0.05% Fast Green (Sigma) and injected into the lateral ventricle of embryos (0.5 µL per embryo) using a beveled glass micropipette. Five electrical pulses (duration: 50 ms) were applied at 42 V across the uterine wall with a 950 ms interval between pulses. Electroporation was performed using a pair of 5 mm platinum electrodes (BTX, Tweezertrode 45-0488, Harvard Apparatus) connected to an electroporator (BTX, ECM830). The embryos were analyzed at E18.5 and P10.

## Immunofluorescence

The sections were first washed with 0.05 M TBS for 10 min, incubated in Triton X-100 (0.5% in 0.05 M TBS) for 30 min at room temperature (RT), and then incubated with blocking solution (5% donkey serum+0.5% Triton X-100 in 0.05 M TBS, pH = 7.2) for 2 hr at RT. The primary antibodies were diluted in 5% donkey serum, incubated overnight at 4°C, and then rinsed three times with 0.05 M TBS.

Secondary antibodies (1:500, all from Jackson ImmunoResearch) matching the appropriate species were added and incubated for 2–4 hr at RT. Fluorescently stained sections were then washed three times with 0.05 M TBS for 10 min. The sections were then stained with 4',6-diamidino-2-phenylindole (1:3000, Sigma) diluted in TBS for 1 min and then finally rinsed three more times with TBS. The sections were then coverslipped with Gel/Mount (Biomeda, Foster City, CA, USA). The primary antibodies used in this study are shown in the Key resources table.

## In situ hybridization

ISH was performed on 20 µm cryosections using digoxigenin riboprobes as previously described (*Yang et al., 2021*). The sections were first postfixed in 4% PFA for 20 min. mRNA in situ hybridization was performed as previously described (*Yang et al., 2021*). The probes were constructed from cDNA isolated from the wild-type mouse brain and amplified by PCR using the following primers:

| Probe name | Primer sequence |
| --- | --- |
| | Primer F: TGGCCAGCAATGTCTCAAAT |
| *Pdgfra* | Primer R: TCCTTTCAAGCATGGGGACA |

## 3D reconstruction

Images of serial coronal sections along the rostral-caudal axis were collected via an Olympus VS 120 microscope using a ×10 objective and used by Amira as a formwork for reconstruction. The coordinates of each labeled cell (tdT+) in the brain were determined. A stereotaxic mouse brain atlas was used to locate the brain anatomical structures where the labeled cells were positioned.

## Image acquisition and statistical analysis

Images for quantitative analyses were acquired with an Olympus VS 120 microscope using a ×10 objective. Bright-field images were acquired with an Olympus VS 120 microscope using a ×10 objective. Other fluorescence images were taken with the Olympus FV3000 confocal microscope system using ×20 or ×40 objectives. The images were merged, cropped, and optimized in Adobe Photoshop and Adobe Illustrator without distorting the original visual information.

Analyses were performed using Microsoft Excel. Unpaired t-test was used to determine statistical significance. All quantitative data are presented as the mean ± SEM. Differences with p-values<0.05 were considered significant. For quantification of SOX10+, PDGFRA+, and GFP+ cells in the cortex at E18.5 and P10, six anatomically matched 20 µm thick coronal sections were selected (n=6 mice per group). We counted SOX10+, PDGFRA+, and GFP+ cells in the cortex under a ×20 objective lens. The data are presented as the number of SOX10+, PDGFRA+, and GFP+ cells per section for each cortex.

For quantification of SOX10+ and PDGFRA+ cells in the dorsal cortex at E14.5 and E16.5, five or six anatomically matched 20 µm thick coronal sections were selected (n=6, E14.5; n=5, E16.5). We

counted SOX10$^+$ and PDGFRA$^+$ cells in the cortex under a ×20 objective lens. The data are presented as the number of SOX10$^+$ and PDGFRA$^+$ cells per section for each cortex.

## Acknowledgements

This work was supported by the Ministry of Science and Technology of China (STI2030-2021ZD0202300), National Natural Science Foundation of China (NSFC 31820103006, 82271197, 81974175, and 32200776), Shanghai Municipal Science and Technology Major Project (No. 2018SHZDZX01), ZJ Lab, and Shanghai Center for Brain Science and Brain-Inspired Technology.

## Additional information

### Funding

| Funder | Grant reference number | Author |
| --- | --- | --- |
| Ministry of Science and Technology of the People's Republic of China | STI2030-2021ZD0202300 | Zhengang Yang |
| National Natural Science Foundation of China | 31820103006 | Zhengang Yang |
| National Natural Science Foundation of China | 82271197 | Dashi Qi |
| National Natural Science Foundation of China | 81974175 | Dashi Qi |
| National Natural Science Foundation of China | 32200776 | Zhuangzhi Zhang |
| Science and Technology Commission of Shanghai Municipality | 2018SHZDZX01 | Zhengang Yang |
| Shanghai Municipal Science and Technology Major Project | ZJ Lab | Zhengang Yang |

The funders had no role in study design, data collection and interpretation, or the decision to submit the work for publication.

### Author contributions

Jialin Li, Formal analysis, Methodology, Project administration; Feihong Yang, Formal analysis, Investigation, Methodology, Project administration; Yu Tian, Software, Investigation, Methodology, Project administration; Ziwu Wang, Xin Wang, Formal analysis, Methodology; Dashi Qi, Resources, Methodology; Zhengang Yang, Conceptualization, Funding acquisition, Writing - review and editing; Jiangang Song, Methodology; Jing Ding, Supervision; Zhuangzhi Zhang, Conceptualization, Formal analysis, Funding acquisition, Writing - original draft, Writing - review and editing

### Author ORCIDs

Zhengang Yang https://orcid.org/0000-0003-2447-6540
Zhuangzhi Zhang https://orcid.org/0000-0002-9860-6689

### Ethics

All experiments conducted in this study were in accordance with the guidelines of Fudan University (No.20220228-140).

Reviewer #1 (Public Review): https://doi.org/10.7554/eLife.94317.3.sa1
Reviewer #2 (Public Review): https://doi.org/10.7554/eLife.94317.3.sa2
Author response https://doi.org/10.7554/eLife.94317.3.sa3

## Additional files

### Supplementary files
• MDAR checklist

### Data availability
All data generated or analyzed during this study are included in the manuscript and supporting files.

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
