## [Editor Report · eLife assessment]

The authors provide **solid** evidence that any contribution of oligodendrocyte precursors to the developing cortex from the lateral ganglionic eminence is minimal in scope. The methods used support the conclusions, with some technical concerns that the authors can address with further experimentation. These are considered **valuable** additions to our understanding of the origins of oligodendrocytes in the forebrain during development.

---

## [Referee Report · Reviewer #1 (Public Review)]

Summary:

In this manuscript the authors re-examine the developmental origin of cortical oligodendrocyte (OL) lineage cells using a combination of strategies, focussing on the question of whether the LGE generates cortical OL cells. The paper is interesting to myelin biologists, the methods used are appropriate and, in general, the study is well-executed, thorough, and persuasive, but not 100% convincing.

Strengths, weaknesses, and recommendations:

The first evidence presented that the LGE does not generate OLs for the cortex is that there are no OL precursors 'streaming' from the LGE during embryogenesis, unlike the MGE (Figure 1A). This in itself is not strong evidence, as they might be more dispersed. In fact, in the images shown, there is no obvious 'streaming' from the MGE either. Note that in Figure 1 there is no reference to the star that is shown in the figure.

The authors then electroporate a reporter into the LGE at E13.5 and examine the fate of the electroporated cells (Figures 1C-E). They find that electroporated cells became neurons in the striatum and in the cortex but no OLs for the cortex. There are two issues with this: first, there is no quantification, which means there might indeed be a small contribution from the LGE that is not immediately obvious from snapshot images. Second, it is unexpected to find labelled neurons in the cortex at all since the LGE does not normally generate neurons for the cortex! Electroporations are quite crude experiments as targeting is imprecise and variable and not always discernible at later stages. For example, in Figure 1D, one can see tdTOM+ cells near the AEP, as well as the striatum. Hence, IUE cannot on its own be taken as proof that there is no contribution of the LGE to the cortical OL population.

The authors then use an alternative fate-mapping approach, again with E13.5 electroporations (Figure 2). They find only a few GFP+ cells in the cortex at E18 (Figures 2C-D) and P10 (Figure 2E) and these are mainly neurons, not OL lineage cells. Again, there is no quantification.

Figure 3 is more convincing, but the experiments are incomplete. Here the authors generate triple-transgenic mice expressing Cre in the cortex (Emx1-Cre) and the MGE (Nkx2.1-Cre) as well as a strong nuclear reporter (H2B-GFP). They find that at P0 and P10, 97-98% of OL-lineage cells (SOX10+ or PDGFRA+) in the cortex are labelled with GFP (Figure 3). This is a more convincing argument that the LGE/CGE might not contribute significant numbers of OL lineage cells to the cortex, in contrast to the Kessaris et at. (2006) paper, which showed that Gsh2-Cre mice label ~50% of SOX10+ve cells in the motor cortex at P10. The authors of the present paper suggest that the discrepancy between their study and that of Kessaris et al. (2006) is based on the authors' previous observation (Zhang et al 2020) (https://doi.org/10.1016/j.celrep.2020.03.027) that GSH2 is expressed in intermediate precursors of the cortex from E18 onwards. If correct, then Kessaris et al. might have mistakenly attributed Gsh2-Cre+ lineages to the LGE/CGE when they were in fact intrinsic to the cortex. However, the evidence from Zhang et al 2020 that GSH2 is expressed by cortical intermediate precursors seems to rest solely on their location within the developing cortex; a more convincing demonstration would be to show that the GSH2+ putative cortical precursors co-label for EMX1 (by immunohistochemistry or in situ hybridization), or that they co-label with a reporter in Emx1-driven reporter mice. This demonstration should be simple for the authors as they have all the necessary reagents to hand. Without these additional data, the assertion that GSX2+ve cells in the cortex are derived from the cortical VZ relies partly on an act of faith on the part of the reader.

Note that Tripathi et al. (2011, "Dorsally- and ventrally-derived oligodendrocytes have similar electrical properties but myelinate preferred tracts." J. Neurosci. 31, 6809-6819) found that the Gsh-Cre+ OL lineage contributed only ~20% of OLs to the mature cortex, not ~50% as reported by Kessaris et al. (2006). If it is correct that these Gsh2-derived OLs are from the cortical anlagen as the current paper claims, then it would raise the possibility that the ventricular precursors of GSH2+ intermediate progenitors are not uniformly distributed through the cortical VZ but are perhaps localized to some part of it. Then the contribution of Gsh2-derived OLs to the cortical population could depend on precisely where one looks relative to that localized source. It would be a nice addition to the current manuscript if the authors could explore the distribution of their GSH2+ intermediate precursors throughout the developing cortex. In any case, Tripathi et al. (2011) should be cited.

Finally, the authors deleted Olig2 in the MGE and found a dramatic reduction of PDGFRA+ and SOX10+ cells in the cortex at E14 and E16 (Figure 4A-F). This further supports their conclusion that, at least at E16, there is no significant contribution of OLs from ventral sources other than the MGE/AEP. This does not exclude the possibility that the LGE/CGE generates OLs for the cortex at later stages. Hence, on its own, this is not completely convincing evidence that the LGE generates no OL lineage cells for the cortex.

Comments on the latest version:

The revised manuscript has addressed the issues we raised previously. The addition of the new Figure 3 supplement 1A-C demonstrating that Gsx2+ve cells in the cortex are generated from Emx1-Cre precursors is convincing, although there is nothing to prove that the GFP+, Gsh2+ double-labelled nuclei are oligodendrocyte lineage and not, for example, astrocytes. It would be helpful to include a Gsh2, Olig2 (or Gsh2, Sox10) double-label image to prove this point. Also, to make the figure more clear, the authors should also show a small area at high magnification, splitting the green and red channels so that the reader can see more clearly that all the red cells are also green.

---

## [Referee Report · Reviewer #2 (Public Review)]

Traditional thinking has been that cortical oligodendrocyte progenitor cells (OPCs) arise in the development of the brain from the medial ganglionic eminence (MGE), lateral/caudal ganglionic eminence (LGE/CGE), and cortical radial glial cells (RGCs). Indeed a landmark study demonstrated some time ago that cortical OPCs are generated in three waves, starting with a ventral wave derived from the medial ganglionic eminence (MGE) or the anterior entopeduncular area (AEP) at embryonic day E12.5 (Nkx2.1+ lineage), followed by a second wave of cortical OLs derived from the lateral/caudal ganglionic eminences (LGE/CGE) at E15.5 (Gsx2+/Nkx2.1- lineage), and then a final wave occurring at P0, when OPCs originate from cortical glial progenitor cells (Emx1+ lineage). However, the authors challenge the idea in this paper that cortical progenitors are produced from the LGE. They have found previously that cortical glial progenitor cells were also found to express Gsx2, suggesting this may not have been the best marker for LGE-derived OPCs. They have used fate mapping experiments and lineage analyses to suggest that cortical OPCs do not derive from the LGE.

Strengths:

(1) The data is high quality and very well presented, and experiments are thoughtful and elegant to address the questions being raised.

(2) The authors use two elegant approaches to lineage trace LGE derived cells, namely fate mapping of LGE-derived OPCs by combining IUE (intrauterine electroporation) with a Cre recombinase-dependent IS reporter, and Lineage tracing of LGE-derived OPCs by combining IUE with the PiggyBac transposon system. Both approaches show convincingly that labelled LGE-derived cells that enter the cortex do not express OPC markers, but that those co-labelling with oligodendrocyte markers remain in the striatum.

(3) The authors then use further approaches to confirm their findings. Firstly they lineage trace Emx1-Cre; Nkx2.1-Cre; H2B-GFP mice. Emx1-Cre is expressed in cortical RGCs and Nkx2.1-Cre is specifically expressed in MGE/AEP RGCs. They find that close to 98% of OPCs in the cortex co-label with GFP at later times, suggesting the contribution of OPCs from LGE is minimal.

(4) They use one further approach to strengthen the findings yet further. They cross Nkx2.1-Cre mice with Olig2 F/+ mice to eliminate Olig2 expression in the SVZ/VZ of the MGE/AEP (Figures 4A-B). The generation of MGE/AEP-derived OPCs is inhibited in these Olig2-NCKO conditional mice. They find that the number of cortical progenitors at E16.5 is reduced 10-fold in these mice, suggesting that LGE contribution to cortical OPCs is minimal.

Impact of Study:

The authors show elegantly and convincingly that the contribution of the LGE to the pool of cortical OPCs is minimal. The title should perhaps be that the LGE contribution is minimal rather than no contribution at all, as they are not able to rule out some small contribution from the LGE. These findings challenge the traditional belief that the LGE contributes to the pool of cortical OPCs. The authors do show that the LGE does produce OPCs, but that they tend to remain in the striatum rather than migrate into the cortex. It is interesting to wonder why their migration patterns may be different from the MGE-derived OPCs which migrate to the cortex. The functional significance of these different sources of OPCs for adult cortex in homeostatic or disease states remains unclear though.

---

## [Author Response]

The following is the authors’ response to the original reviews.

**Reviewer #1 (Public Review):**
Summary:In this manuscript the authors re-examine the developmental origin of cortical oligodendrocyte (OL) lineage cells using a combination of strategies, focussing on the question of whether the LGE generates cortical OL cells. The paper is interesting to myelin biologists, the methods used are appropriate and, in general, the study is well-executed, thorough, and persuasive, but not 100% convincing.

Thank you very much for approving our paper.

Strengths, weaknesses, and recommendations:The first evidence presented that the LGE does not generate OLs for the cortex is that there are no OL precursors 'streaming' from the LGE during embryogenesis, unlike the MGE (Figure 1A). This in itself is not strong evidence, as they might be more dispersed. In fact, in the images shown, there is no obvious 'streaming' from the MGE either. Note that in Figure 1 there is no reference to the star that is shown in the figure.

We totally agree with you. While OPC migration stream is not strong evidence to support that the LGE does not generate OPCs for the cortex, when considering our additional evidence, the absence of obvious 'streaming' from LGE to cortex provided supplementary support for this conclusion. Finally, we have removed the star in the figure.

The authors then electroporate a reporter into the LGE at E13.5 and examine the fate of the electroporated cells (Figures 1C-E). They find that electroporated cells became neurons in the striatum and in the cortex but no OLs for the cortex. There are two issues with this: first, there is no quantification, which means there might indeed be a small contribution from the LGE that is not immediately obvious from snapshot images. Second, it is unexpected to find labelled neurons in the cortex at all since the LGE does not normally generate neurons for the cortex. Electroporations are quite crude experiments as targeting is imprecise and variable and not always discernible at later stages. For example, in Figure 1D, one can see tdTOM+ cells near the AEP, as well as the striatum. Hence, IUE cannot on its own be taken as proof that there is no contribution of the LGE to the cortical OL population.

Thank you for your constructive suggestions.

(1) Following the reviewer's suggestion, we have added these statistics, please see Figure 1F.

(2) The reviewer raised a good point. We occasionally found a very small number of electroporated cells in the MGE/AEP VZ in our IUE system. Therefore, we can identify these electroporated cells in the cortex, most of them expressed the neuronal marker NeuN. We suspect these are MGE-derived cortical interneurons. It's worth noting that these electroporated cells (MGE-derived) are not glia cells. The probable reason may be that MGE/AEP generate cortical OPCs mainly before E13.5 (in this study we performed IUE at E13.5).

The authors then use an alternative fate-mapping approach, again with E13.5 electroporations (Figure 2). They find only a few GFP+ cells in the cortex at E18 (Figures 2C-D) and P10 (Figure 2E) and these are mainly neurons, not OL lineage cells. Again, there is no quantification.

Thank you very much for your suggestions. Actually, in this fate-mapping approach, the electroporated cells in the cortex is very few. We analyzed four mice, and found that all GFP positive cells (139 GFP+) did not express OLIG2, SOX10 and PDGFRA.

Figure 3 is more convincing, but the experiments are incomplete. Here the authors generate triple-transgenic mice expressing Cre in the cortex (Emx1-Cre) and the MGE (Nkx2.1-Cre) as well as a strong nuclear reporter (H2B-GFP). They find that at P0 and P10, 97-98% of OL-lineage cells (SOX10+ or PDGFRA+) in the cortex are labelled with GFP (Figure 3). This is a more convincing argument that the LGE/CGE might not contribute significant numbers of OL lineage cells to the cortex, in contrast to the Kessaris et at. (2006) paper, which showed that Gsh2-Cre mice label ~50% of SOX10+ve cells in the motor cortex at P10. The authors of the present paper suggest that the discrepancy between their study and that of Kessaris et al. (2006) is based on the authors' previous observation (Zhang et al 2020) (https://doi.org/10.1016/j.celrep.2020.03.027) that GSH2 is expressed in intermediate precursors of the cortex from E18 onwards. If correct, then Kessaris et al. might have mistakenly attributed Gsh2-Cre+ lineages to the LGE/CGE when they were in fact intrinsic to the cortex. However, the evidence from Zhang et al 2020 that GSH2 is expressed by cortical intermediate precursors seems to rest solely on their location within the developing cortex; a more convincing demonstration would be to show that the GSH2+ putative cortical precursors co-label for EMX1 (by immunohistochemistry or in situ hybridization), or that they co-label with a reporter in Emx1-driven reporter mice. This demonstration should be simple for the authors as they have all the necessary reagents to hand. Without these additional data, the assertion that GSX2+ve cells in the cortex are derived from the cortical VZ relies partly on an act of faith on the part of the reader. Note that Tripathi et al. (2011, "Dorsally- and ventrally-derived oligodendrocytes have similar electrical properties but myelinate preferred tracts." J. Neurosci. 31, 6809-6819) found that the Gsh-Cre+ OL lineage contributed only ~20% of OLs to the mature cortex, not ~50% as reported by Kessaris et al. (2006). If it is correct that these Gsh2-derived OLs are from the cortical anlagen as the current paper claims, then it would raise the possibility that the ventricular precursors of GSH2+ intermediate progenitors are not uniformly distributed through the cortical VZ but are perhaps localized to some part of it. Then the contribution of Gsh2-derived OLs to the cortical population could depend on precisely where one looks relative to that localized source. It would be a nice addition to the current manuscript if the authors could explore the distribution of their GSH2+ intermediate precursors throughout the developing cortex. In any case, Tripathi et al. (2011) should be cited.

Thank you for your constructive suggestions.

(1) We used the Emx1Cre; RosaH2B-GFP mouse and found that nearly all GSX2+ cells in the cortical SVZ are derived from the Emx1+ lineage at P0 (Please see our new Figure 3-supplement 1A-C).

(2) According to your suggestion, we have cited this paper (Tripathi et al.) in our revised manuscript.

(3) The study conducted by Kessaris et al. (2006) revealed that roughly 50% of cortical oligodendrocytes (OLs) originate from the Gsx2 lineage (LGE/CGE-derived). In contrast, Tripathi et al. (2011) observed that Gsx2-derived OLs contribute only around 20% to the corpus callosum (CC). To investigate the reasons behind these disparate findings, we conducted three experiments. Firstly, using Emx1Cre; RosaH2B-GFP mice, we found that approximately 89% of lateral CC (LCC) OLs originate from the Emx1 lineage, with only around 11% derived from the ventral source (refer to Author response image 1A and B below). Secondly, employing Nkx2-1Cre; RosaH2B-GFP mice, we determined that approximately 11% of LCC OLs originate from the Nkx2.1 lineage (refer to pictures C and D below). Finally, we found that approximately 98.3% of lateral LCC OLs originate from both Emx1 and Nkx2.1 lineages, with only around 1.7% possibly derived from the LGE (see Author response image 1E and F below). Taken together, our results indicate that approximately 89% of LCC OLs originate from the Emx1 lineage, while 11% of LCC OLs are derived from the medial ganglionic eminence (MGE).

It is worth noting that OLs from Emx1 and Nkx2.1 lineages were equally distributed in the medial CC (mCC) (see Author response image 1G below). This finding suggests that MGE-derived OLs exhibit spatial heterogeneity in their distribution within the CC. These results provide evidence that the contribution of the lateral ganglionic eminence (LGE) and caudal ganglionic eminence (CGE) to CC OLs is minimal.

Finally, the authors deleted Olig2 in the MGE and found a dramatic reduction of PDGFRA+ and SOX10+ cells in the cortex at E14 and E16 (Figure 4A-F). This further supports their conclusion that, at least at E16, there is no significant contribution of OLs from ventral sources other than the MGE/AEP. This does not exclude the possibility that the LGE/CGE generates OLs for the cortex at later stages. Hence, on its own, this is not completely convincing evidence that the LGE generates no OL lineage cells for the cortex.

There are three reasons why we didn't analyze *Olig2*-NCKO mice after E16.5. 1. The expression of Nkx2.1Cre is lower within the dorsal-most region of the MGE than other Nkx2.1-expressing regions. Even at E15.5, we can still find a small number of OPCs in the lateral cortex. We speculate that these OPCs are derived from dorsal MGE. 2. Considering the possibility of incomplete recombination in Olig2 gene locus, we guess OPCs (Olig2+) in the lateral cortex are derived from MGE. Indeed, we found a few OPCs in the MGE/AEP in the *Olig2*-NCKO mice (Figure 4F). 3. The recent study (bioRxiv preprint doi: https://doi.org/10.1101/2024.01.23.576886) showed that the contribution of LGE/CGE to cortical OPCs is minimal, which further supporting our findings. Taken together, our results provide additional evidence supporting the limited contribution of the LGE/CGE to cortical OPCs (OLs).

**Reviewer #2 (Public Review):**
Traditional thinking has been that cortical oligodendrocyte progenitor cells (OPCs) arise in the development of the brain from the medial ganglionic eminence (MGE), lateral/caudal ganglionic eminence (LGE/CGE), and cortical radial glial cells (RGCs). Indeed a landmark study demonstrated some time ago that cortical OPCs are generated in three waves, starting with a ventral wave derived from the medial ganglionic eminence (MGE) or the anterior entopeduncular area (AEP) at embryonic day E12.5 (Nkx2.1+ lineage), followed by a second wave of cortical OLs derived from the lateral/caudal ganglionic eminences (LGE/CGE) at E15.5 (Gsx2+/Nkx2.1- lineage), and then a final wave occurring at P0, when OPCs originate from cortical glial progenitor cells (Emx1+ lineage). However, the authors challenge the idea in this paper that cortical progenitors are produced from the LGE. They have found previously that cortical glial progenitor cells were also found to express Gsx2, suggesting this may not have been the best marker for LGE-derived OPCs. They have used fate mapping experiments and lineage analyses to suggest that cortical OPCs do not derive from the LGE.Strengths:(1) The data is high quality and very well presented, and experiments are thoughtful and elegant to address the questions being raised.(2) The authors use two elegant approaches to lineage trace LGE derived cells, namely fate mapping of LGE-derived OPCs by combining IUE (intrauterine electroporation) with a Cre recombinase-dependent IS reporter, and Lineage tracing of LGE-derived OPCs by combining IUE with the PiggyBac transposon system. Both approaches show convincingly that labelled LGE-derived cells that enter the cortex do not express OPC markers, but that those co-labelling with oligodendrocyte markers remain in the striatum.(3) The authors then use further approaches to confirm their findings. Firstly they lineage trace Emx1-Cre; Nkx2.1-Cre; H2B-GFP mice. Emx1-Cre is expressed in cortical RGCs and Nkx2.1-Cre is specifically expressed in MGE/AEP RGCs. They find that close to 98% of OPCs in the cortex co-label with GFP at later times, suggesting the contribution of OPCs from LGE is minimal.(4) They use one further approach to strengthen the findings yet further. They cross Nkx2.1-Cre mice with Olig2 F/+ mice to eliminate Olig2 expression in the SVZ/VZ of the MGE/AEP (Figures 4A-B). The generation of MGE/AEP-derived OPCs is inhibited in these Olig2-NCKO conditional mice. They find that the number of cortical progenitors at E16.5 is reduced 10-fold in these mice, suggesting that LGE contribution to cortical OPCs is minimal.

We thank the reviewer for summarizing the strengths of our manuscript.

Weaknesses:(1) The authors use IUE in experiments mentioned in point 2 of 'Strengths' above (Figures 1 and 2) and claim that the reporter was delivered specifically into LGE VZ at E13.5 using this IUE. It would be nice to see some sort of time course of delivery after IUE to show the reporter is limited to LGE VZ at early times post-IUE.

Thank you very much for your suggestions. Indeed, when using IUE in our system, we occasionally found a small number of electroporated cells in the MGE/AEP VZ. Thus, we can find very few electroporated cells (MGE/AEP-derived) in the cortex and these electroporated cells are neuron (perhaps interneuron).

(2) In the experiments mentioned in point 3 of 'Strengths' (Figure 3), statistical analysis showed that only approximately 2% of OPCs were GFP-negative cells. This 2% could possibly be derived from the LGE/CGE so does not totally rule out that LGE contributes some cortical OPCs.

Thank you for your constructive suggestions. We apologize for any imprecise descriptions. Despite we suspect that this 2% may originate from MGE {Considering the possibility of incomplete recombination in Olig2 gene locus, we guess the OPCs (Olig2+) may be derived from MGE. Indeed, we found a few OPCs in the MGE/AEP in the *Olig2*-NCKO mice (Figure 4F)} or from the dMGE (The expression of Nkx2.1Cre is lower within the dorsal-most region of the MGE than in other Nkx2.1-expressing regions). Anyway, we have softened the assertion everywhere in our revised manuscript.

(3) In the experiments mentioned in point 4 of 'Strengths' (Figure 4), they do still find cortical OPCs at E16.5 in the Olig2-NCKO conditional mice. It is unclear whether this is due to the recombination efficiency of the CRE enzyme not being 100%, or whether there is some LGE contribution to the cortical OPCs.

This experiment alone may not provide strong evidence to support that LGE do not contribute to the cortical OPCs during development. However, when combing our other results with this result, we can confirm that the contribution of LGE to cortical OPCs is minimal. Furthermore, a recent study reported that LGE/CGE-derived OLs make minimum contributions to the neocortex and corpus callosum，which further supporting the reliability of our conclusion.

We would like to thank the reviewers and editors for their valuable comments and suggestions again.

Impact of Study:The authors show elegantly and convincingly that the contribution of the LGE to the pool of cortical OPCs is minimal. The title should perhaps be that the LGE contribution is minimal rather than no contribution at all, as they are not able to rule out some small contribution from the LGE. These findings challenge the traditional belief that the LGE contributes to the pool of cortical OPCs. The authors do show that the LGE does produce OPCs, but that they tend to remain in the striatum rather than migrate into the cortex. It is interesting to wonder why their migration patterns may be different from the MGE-derived OPCs which migrate to the cortex. The functional significance of these different sources of OPCs for adult cortex in homeostatic or disease states remains unclear though.
**Recommendations for the authors:**

**Reviewer #1 (Recommendations for The Authors):**
(1) Change the title to e.g. 'limited contribution of the LGE to cortical oligodendrocytes'. Alternatively, It might be more useful to highlight where they come from, e.g. "Cortical oligodendrocytes originate predominantly or exclusively from the MGE and cortical VZ"

As suggested, we have changed the old title to the following: The lateral/caudal ganglionic eminence makes a limited contribution to cortical oligodendrocytes

(2) Demonstrate using lineage tracing that GSH2+ cells in the cortex are derived from the Emx1-lineage, e.g. using immunohistochemistry for GSX2 and a reporter in Emx1-Cre mice crossed to a reporter.

In our revised manuscript, we have added a new figure (Figure 3-supplement 1A-C) to demonstrate that the GSX2+ cells in the cortex are derived from the Emx1-lineage.

(3) Make it clear in their discussion that they have not explored the CGE so it is possible that this region generates some OLs.

The Emx1Cre; Nkx2.1Cre; H2B-GFP mice showed that only ~2% cortical OLs are derived from LGE/CGE. Actually, considering the efficiency of Cre enzyme recombination and the relatively low Cre activity in the dMGE of Nkx2.1Cre, the actual contribution of LGE/CGE-derived cortical OLs could be even lower than our current observation. Therefore, our results demonstrate that the LGE/CGE generate very few，possibly even no，OLs for the cortex.

(4) Soften the assertion that the LGE does not generate any OL lineage cells that reach the cortex by e.g. changing the word 'sole' to 'predominant' (line 88) and, elsewhere in the paper, leaving open the possibility that small numbers of LGE-derived OLs might enter the cortex, depending on where exactly one looks.

As suggested, we have softened the assertion everywhere in our manuscript.

(5) Lines 255-260: 'First, the time window during which the MGE generates OLs is very brief, perhaps occurring before MGE neurogenesis. The high level of SHH in the MGE allows for the production of a small population of cortical OPCs around E12.5. Subsequently, multipotent intermediate progenitors begin to express DLX transcription factors resulting in ending the generation of OPCs in the MGE'. What is the evidence that OL genesis precedes neurogenesis? If there is none (as I suspect) then this statement should be removed.

The editors raised a good point. We have no strong evidence to support that OL genesis precedes neurogenesis in MGE, thus, we removed these sentences in our manuscript.

(6) Figure 1E should show quantification of cells as a % of electroporated cells and as a % of PDGFRA+ or OLIG2+ or SOX10+ cells, so that the reader might have a clear view of the extent of labelling.

Done.

(7) Figure 4: This is interesting but incomplete. At E14.5 the authors show the presence of PDGFRA+ cells in the telencephalon. However, at E16.5 they show images only of the dorsal-most region of the cortex. If the LGE/CGE begins to generate OLPs for the early cortex, they would be expected to appear near the cortico-striatal boundary, as shown in Kessaris 2006 Fig1g-h. In the current manuscript, the authors do not show these regions, or the LGE and CGE, in their images. It is essential to show PDGFRA immunolabelling at the cortico-striatal boundary and also in the LGE and CGE at E16.5 in control and Olig2 mutant mice. It is also necessary to extend this analysis to E18.5, perhaps showing PDGFRA + cells streaming from the cortical VZ/SVZ.

There are three reasons why we didn't analyze *Olig2*-NCKO mice after E16.5. 1.Frankly, the expression of Nkx2.1Cre is lower within the dorsal-most region of the MGE than other Nkx2.1-expressing regions. Even at E15.5, we can still find a small number of OPCs in the lateral cortex. We guess these OPCs are derived from dMGE. 2. Considering the possibility of incomplete recombination in Olig2 gene locus, we guess OPCs (Olig2+) are derived from MGE. In fact, we found a few OPCs in the MGE/AEP in the *Olig2*-NCKO mice (Figure 4F). 3. The recent study (bioRxiv preprint doi: https://doi.org/10.1101/2024.01.23.576886) showed that the contribution of LGE/CGE to cortical OPCs is minimal. Taken together, our results provide additional evidence supporting the limited contribution of the LGE/CGE to cortical OPCs (OLs).

(8) Cite Tripathi et al. (2011) and mention the disparity between the findings of that paper and Kessaris et al. (2006) and possible reasons - see main review above.

Done.